# Get a Grip: Multi-Finger Grasp Evaluation at Scale Enables Robust Sim-to-Real Transfer

**Tyler Ga Wei Lum**[*,†]**, Albert H. Li**[*,‡]**, Preston Culbertson**[‡]**, Krishnan Srinivasan**[†]**,**
**Aaron D. Ames**[‡]**, Mac Schwager**[†]**, Jeannette Bohg**[†]
[*] Equal Contribution, [†] Stanford University, [‡] Caltech
{tylerlum, krshna, schwager, bohg}@stanford.edu
{alberthli, pculbert, ames}@caltech.edu

**Abstract:** This work explores conditions under which multi-finger grasping algorithms can attain robust sim-to-real transfer. While numerous large datasets facilitate learning *generative* models for multi-finger grasping at scale, reliable real-world dexterous grasping remains challenging, with most methods degrading when deployed on hardware. An alternate strategy is to use *discriminative* grasp evaluation models for grasp selection and refinement, conditioned on real-world sensor measurements. This paradigm has produced state-of-the-art results for vision-based parallel-jaw grasping, but remains underutilized in the multi-finger setting. In this work, we contend that existing datasets and methods have been insufficient for training performant discriminative models for multi-finger grasping. To train grasp evaluators at scale, datasets must provide on the order of millions of grasps, including both positive *and negative* examples, with corresponding perceptual data resembling measurements at inference time. To that end, we release a new, open-source dataset of 3.5M grasps on 4.3K objects annotated with RGB images, point clouds, and trained NeRFs. Leveraging this dataset, we train multiple vision-based grasp evaluators that outperform both analytic and generative modeling-based baselines without evaluators on extensive simulated and real-world trials across a diverse range of objects. We show via numerous ablations that the key factor for performance is indeed the evaluators, and that their quality degrades as the dataset shrinks, demonstrating the importance of our new dataset. Project website at: https://sites.google.com/view/get-a-grip-dataset.

**Keywords:** Multi-Fingered Grasping, Large-Scale Grasp Dataset, Sim-to-Real

## 1 Introduction

Dexterous, multi-finger grasping is a longstanding challenge in robotics, with applications in tool use, human-robot collaboration, and robot surgery [1, 2, 3]. A common approach for grasp synthesis samples grasp candidates from a generative *sampler* and refines them with a discriminative *evaluator* using some grasp metric. Table 1 summarizes prior works employing this "coarse-to-fine" approach.

In terms of grasp evaluation, classical methods [4, 5, 6] use *analytic* metrics [7] to optimize grasps based on grasp mechanics, which require ground-truth object geometry and are sensitive to sensor noise [8, 9]. In contrast, *data-driven* grasp evaluators [10, 11, 12] estimate grasp quality by conditioning on realistic perceptual data, thus better predicting empirical robustness. While this discriminative approach to grasp synthesis has proven effective for parallel-jaw grasping [13, 14, 15, 16], its application to the multi-finger setting has seen considerably less success. Consequently, many works study an alternate generative approach of learning and sampling from a distribution over high-quality grasps using large-scale multi-finger grasp datasets [17, 18, 19, 20, 21]. However, the real-world uncertainties in modeling and perception introduce a significant sim-to-real gap that complicates reliable grasp generation on hardware. Therefore, this work's main goal is the synthesis of

8th Conference on Robot Learning (CoRL 2024), Munich, Germany.

Table 1: **Comparison of dexterous grasp synthesis methods.** SDF = Signed Distance Field, BPS = Basis Point Set, CVAE = Conditional Variational Autoencoder, EBM = Energy-Based Model.

| Method | Sampler | Evaluator | Obj. Data |
|---|---|---|---|
| GraspIt! [4] | Heuristic | Analytic (Ferrari-Canny) | Shape primitives |
| FRoGGeR [22] | Heuristic | Analytic (Min-weight) | Mesh, SDF |
| Wu et al. [23] | Learned (CVAE) | Analytic (Force closure) | Point cloud |
| DexGraspNet [17] | Heuristic | Analytic (Force closure proxy) | Mesh |
| Lu et al. [10] | Heuristic | Learned (CNN) | RGB-D |
| FFHNet [24] | Learned (CVAE) | Learned (Classifier) | BPS |
| DexDiffuser [25] | Learned (Diffusion) | Learned (Classifier) | BPS |
| UniDexGrasp [18] | Learned (EBM) | Learned (RL) | Point cloud |
| Ours | - | Learned (Success prob.) | BPS, NeRF |

robust, real-world dexterous grasps on novel objects, a challenge that has persisted despite numerous previous efforts.

Upon re-examining the success of discriminative evaluator-based approaches for parallel-jaw grasp synthesis, we observed 3 key criteria in the associated training data: (i) positive *and negative* grasp examples with high-quality labels, (ii) corresponding perceptual data (e.g., point clouds) used to train measurement-conditioned models, and (iii) a large enough scale to allow generalization in the real world. For example, the evaluator in the seminal work DexNet 2.0 [14] was trained on over 6.7M triples of simulated parallel-jaw grasps, top-down images of the workspace, and expected grasp quality labels (aggregated over several perturbations of each grasp).

In comparison, for multi-finger grasping, no dataset satisfies all three criteria. Almost all recent large datasets only satisfy criteria (i) or (ii), and never simultaneously, because they were designed with generative models in mind (Table 2). Conversely, prior works on evaluator-based methods typically only employ (closed-source) datasets with tens of thousands of grasps, failing to satisfy criterion (iii). We show the historical trend of evaluator dataset sizes in Fig. 18 of Appendix E.1.

In light of this, we hypothesize that like in parallel-jaw grasping, a grasp evaluator trained on a large-scale dataset satisfying the above criteria would enable robust, real-world, multi-finger grasp synthesis. We test this in a controlled setting where we first train various grasp evaluators on a large dataset of purely simulated grasps and perceptual data, then deploy them zero-shot in the real world on a robotic manipulator over a diverse set of objects. Our many experiments and ablations show that evaluators improve performance (compared to purely generative methods [18, 24, 25, 23]), the resulting performance boosts are uniquely attributable to the evaluators, and dataset size is indeed crucial for performance. In summary, our contributions are as follows.

**An open-source, large-scale dataset.** Our dataset consists of 3.5M grasps (for each of the Allegro and LEAP [26] hands) on 4.3K unique objects at multiple scales, which is larger (per-hand) than all existing multi-finger grasp datasets [17, 20, 27]. We release synthetic images of these objects from multiple views, corresponding point clouds, and NeRFs trained over these images. We include over 250 successful and failed grasps for each object and soft labels indicating quantities such as the probability of grasp success (PGS).

**Comprehensive simulated and real-world evaluations.** We test our evaluators trained on our dataset on thousands of simulated and hundreds of real-world grasps on the Allegro hand, demonstrating state-of-the-art performance. We show that grasp evaluators provide a clear, uniquely attributable performance improvement over evaluator-free methods. Further, our work is the first to quantify the positive effect that dataset scale has on grasp success rates in the real world.

## 2    Related Work

Traditional approaches to (precision) dexterous grasp synthesis optimize **analytic metrics** for grasp quality. These methods focus on contacts between robot hand and object, typically optimizing for robustness to external disturbances. For a review of this large body of work, we refer to [28, 29]. In

Table 2: **Dexterous Grasp Dataset Comparison.** PGS = probability of grasp success. (*) denotes total number of grasps divided by number of hands in the dataset.

| Dataset | Grasps | Labels | Objects | Code + Data | Observations | Has Table |
|---|---|---|---|---|---|---|
| MultiDex [27] | ~87k* | +ve only | 58 | Yes | - | Possible |
| Grasp'D-1M [45] | ~333k* | +ve only | 2.3k | No | Visual | Yes |
| DexGraspNet [17] | 1.32M | +ve only | 5.3k | Yes | - | No |
| MultiGripperGrasp [46] | ~2.8M* | Ranked | 345 | Yes | - | No |
| **Get a Grip (Ours)** | 3.5M | PGS | 4.3k | Yes | Visual | Yes |

general, these approaches [6, 4, 5] are initialized using a heuristic sampler and use sampling-based optimization to optimize an analytic grasp metric. More recent approaches [23, 22, 30] leverage bilevel optimization to generate provably force closure grasps. These approaches suffer when accurate models of object geometry are unavailable, which must either be recorded with specialized scanning rigs [31, 32] or reconstructed from visual information into low-quality meshes [30].

When generating grasps in the real world, grasp planners must use **real-world object representations** like RGB images, depth images, or point clouds. A more recent promising representation is the basis point set (BPS) [33], a fixed-length feature vector containing the minimal distances to a fixed set of basis points. Neural Radiance Fields (NeRFs) [34] are another representation used for model view synthesis that has proven useful for grasp generation, representing affordances, and semantic grasping [35, 36, 37, 38, 39, 40, 41]. Importantly, all of these works study only parallel-jaw grippers, which have limited use for complex manipulation tasks. One reason for this emphasis is that they can be parameterized with a single element of SE(3), and it is clear that the relevant NeRF features lie between the jaws. In contrast, the parameterization of multi-finger grasps is not as straightforward; we address this challenge below.

**Data-driven grasp synthesis** methods, as extensively reviewed in [42, 8], leverage grasp datasets to train grasp planners that take real-world perceptual information as input. Many works have used simulation to generate large parallel-jaw grasping datasets [43, 14], which have shown to be highly effective [44, 14]. However, generating large-scale datasets for multi-fingered hands is much more complex due to the high number of degrees of freedom and ambiguities in grasp parameterization. Despite this, recent works have released large-scale dexterous grasping datasets, as shown in Table 2, such as MultiDex [27], DexGraspNet [17], and Grasp'D-1M [45]. One key limitation of these datasets is that they only contain positive examples, so they cannot be used to train evaluator models that can distinguish between successful and unsuccessful grasps. One counterexample is MultiGripperGrasp [46], which uses GraspIt! [4] to generate grasps that are then evaluated in parallel simulation and provides a ranking of grasp goodness. However, it is currently limited to a small set of 345 objects, does not contain visual observations, and does not account for table collisions, which are important for most grasping applications. Many methods use a combination of sampling and evaluator models, such as Mayer et al. [24], Weng et al. [25], who propose learning both to sample and evaluate multi-fingered grasps using basis point sets as input. However, they needed to generate their own dataset of 180K grasps to train these models. Another line of works focuses on learning differentiable grasp evaluators that can optimize grasps through backpropagation [47, 48, 49, 50]. These works also created their own small datasets on the order of 10k grasps across about 100 objects. Our work seeks to aid in future investigations of evaluator-based grasping by supplying a large-scale dataset which mitigates the substantial effort required to generate a new one.

## 3 Get a Grip: Dataset Construction and Grasp Synthesis

### 3.1 Dataset Generation

Our dataset contains objects with associated grasps, RGB images, point clouds, NeRFs, and labels (illustrated in Figure 1). Let $n_j$ and $n_f$ denote the number of hand joints and fingers. We parameterize precision grasps as $\mathcal{G} = (\mathbf{T}_{OH}, \theta, \mathbf{d}_1, \ldots, \mathbf{d}_{n_f})$, where $\mathbf{T}_{OH} \in SE(3)$ is the hand pose relative to

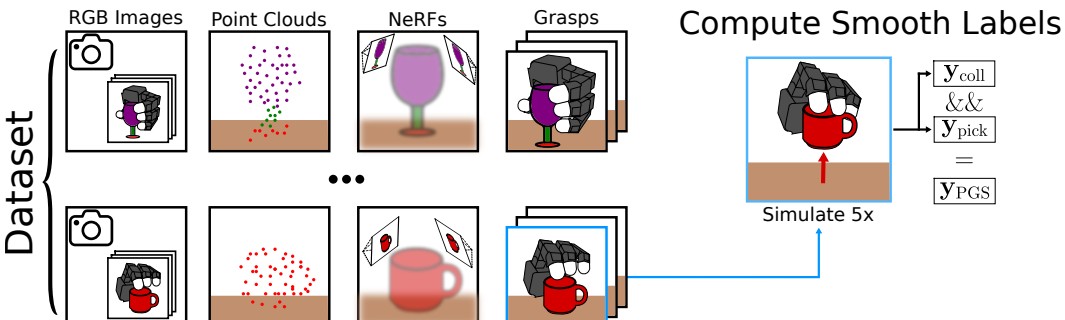

Figure 1: **Data and label generation.** Our dataset supplies the components for robust sim-to-real transfer. For each object in the dataset, we provide RGB images of it from several random views, a full point cloud, and trained NeRF weights, which allows our data to be compatible with most vision-based methods. We also supply hundreds of grasps per object, each of which is simulated in Isaac Gym multiple times with slight wrist pose perturbations. Averaging, this yields three smooth regression targets indicating the probability of (1) unwanted collisions, (2) simulated pick success, and (3) grasp success, the logical conjunction of (2) and (3).

the object, $\theta \in \mathbb{R}^{n_j}$ is the pre-grasp joint configuration, and $\mathbf{d}_i \in S^2$ is the direction in which the $i^{\text{th}}$ fingertip moves during the grasp. The grasp data are stored as tuples $\left\{ \left( \mathcal{G}^k, y_{\text{coll}}^k, y_{\text{pick}}^k, y_{\text{PGS}}^k \right) \right\}$, where each $y_*^k \in [0, 1]$ is a distinct smooth label for grasp $\mathcal{G}^k$ generated in simulation: $y_{\text{coll}}$ denotes a grasp with unwanted collisions, $y_{\text{pick}}$ that the simulated pick was successful, and $y_{\text{PGS}}$ the probability of grasp success, defined as the conjunction of the two.

Our grasp generation pipeline is heavily motivated by Wang et al. [17], but requires key modifications, including (1) placing objects on a table with consistent gravity (instead of floating with different gravity directions) to better match the real world, (2) switching from a binary success label to a continuous success probability label evaluated over multiple trials, (3) modifying evaluation and simulation procedures to improve sim-to-real transfer, and (4) restricting contact points to fingertips.

To procure this dataset, we first generate hundreds of grasps on each object from a diverse object set given by DexGraspNet [17], yielding an initial dataset of about 2M unlabeled grasps. Second, these grasps are simulated in Isaac Gym [51] to compute the associated grasp labels that denote probability of grasp success. Finally, for each of the grasps that are more likely to pass than fail, the dataset is augmented by perturbing each of these grasps several times and re-evaluating them. Because the initial dataset of grasps has many more failed grasps than successful ones, this augmentation gives a new set of grasps with more balanced labels, which, when combined with the initial dataset, yields the final dataset of over 3.5M examples.

**Step 1: Grasp Generation.** We follow the sampling strategy from Wang et al. [17] by defining an energy function $E(\mathcal{G})$ that is the sum of various penalty terms on a grasp and sampling low-energy grasps from the distribution $p(\mathcal{G}) \propto \exp(-E(\mathcal{G}))$ using Langevin sampling. For details, see [17].

The samples are initialized by spawning the hand on some inflated level set of the object mesh such that the palm faces the object with a canonical open-hand joint configuration and then perturbing the configuration with noise. Our weighted energy function (dependence on $\mathcal{G}$ omitted) is

$$E = E_{\text{fc}} + w_{\text{dis}} E_{\text{dis}} + w_{\text{joints}} E_{\text{joints}} + w_{\text{pen}} E_{\text{pen}} + w_{\text{spen}} E_{\text{spen}} + w_{\text{tpen}} E_{\text{tpen}},$$

where $E_{\text{fc}}$ is the differentiable force closure estimator introduced in [17] that encourages force closure grasps; $E_{\text{dis}}$ encourages hand-to-object proximity; $E_{\text{joints}}$ penalizes joint violations; $E_{\text{pen}}$ and $E_{\text{spen}}$ penalize hand-object and self penetration respectively; $E_{\text{tpen}}$ is a new energy term that penalizes hand-table penetration that was added to keep the grasps above the table. Without this term, many optimized grasps would converge to grasps that fully cage the object from above and below in a way that would not be possible in the real world due to table collisions.

Compared to the energy function in DexGraspNet [17], we only specify desired contacts on the fingertips to sample precision grasps, we recover pre-grasp poses with fingertips 1.5cm off the surface instead of on the surface (more clearance aids collision-free motion planning), and we penalize hand-table collisions for kinematic feasibility. Let $\mathbf{p}_{\text{table}}(\cdot)$ return the closest point on the table. Then, the table-penetration energy term is given by

$$E_{\text{tpen}} = \sum_{\mathbf{p} \in P(\mathcal{H})} \max\left((\mathbf{p} - \mathbf{p}_{\text{table}}(\mathbf{p})) \cdot \hat{\mathbf{g}}, 0\right),$$

where $P(\mathcal{H})$ is a set of predefined points on the hand $\mathcal{H}$ and $\hat{\mathbf{g}}$ is the gravity vector. Lastly, for each grasp, the directions $\{\mathbf{d}_i\}$ are the directions of the fingertips to the closest points on the mesh. Specifying these directions eliminates ambiguity in grasp execution, aiding reproducibility.

**Step 2: Grasp Evaluation.** We simulate grasps for the above samples in Isaac Gym. The object is first spawned and allowed to settle, and then the hand is spawned at the pre-grasp pose relative to the object. We assume fingertip $i$ moves by 5cm along $\mathbf{d}_i$, and compute a corresponding closed-hand configuration $\theta_{\text{close}}$ using inverse kinematics. We then use PD control to drive $\theta \to \theta_{\text{close}}$, executing the grasp. Lastly, the object is lifted above the table by 20cm.

For each grasp, we check whether (1) the pre-grasp had no collisions, and (2) it was successfully picked while the hand-object pose did not significantly deviate, yielding labels $y_{\text{coll}}$ and $y_{\text{pick}}$. We need both, as Isaac Gym's physics may exploit nonphysical hand-object penetration to achieve a pick, which we forbid. We then construct the label $y_{\text{PGS}} = y_{\text{coll}} \wedge y_{\text{pick}}$ to indicate the grasp is a true, physically-achievable success. Further, we simulate each grasp 5 times with additional small perturbations, averaging the results to recover non-binary labels in $[0, 1]$ for regression. This smoothing process has been found to help learn robust grasps [52, 53]; in particular, $y_{\text{PGS}}$ is supervised to match an empirical success probability, rather than being a measure of classifier confidence. We emphasize that $y_{\text{coll}}$, $y_{\text{pick}}$, and $y_{\text{PGS}}$ are smooth labels, which are computed by averaging over all perturbations.

In contrast with [17], we check for feasibility in a tabletop setting rather than spawning the scene in a floating environment. We found this assisted sim-to-real transfer by accounting for table penetrations. Moreover, spawning the hand in a pre-grasp (rather than in-contact) position helped eliminate non-physical contact modeling behavior in Isaac Gym that led to instabilities in the simulation labels.

**Step 3: Grasp Augmentation.** For each object, 2-20% of sampled grasps were successes. Small amounts of noise are added to these grasps and fingertip directions 5 times and re-evaluated in Isaac Gym, which yields smoothed, balanced labels after averaging the results. These perturbations are larger than those from Step 2 to promote data diversity. See Appendix B.2 for details.

**Object NeRF Dataset.** In total, we use 4.3K unique meshes, each of which are nominally centered and randomly scaled such that its maximum extent is between 0.1 and 0.3m. To introduce more scale diversity while retaining a dataset size under 2.5TB (see Appendix B.4), a subset of 1700 of the objects are instead duplicated at three distinct scales with max extents drawn uniformly from the distributions $U_{\text{small}}(0.05, 0.1)$, $U_{\text{med}}(0.1, 0.2)$, $U_{\text{large}}(0.2, 0.3)$. Lastly, we train NeRFs of various quality by uniformly sampling 100 images over a spherical cap above the object with radius 0.45m and a polar angle sampled from $U(0, \frac{\pi}{4})$ and training for 400 iterations with `nerfstudio` [54, 55].

## 3.2 Grasp Synthesis with Learned Evaluators

Let the choice of object representation, e.g., a point cloud, be denoted $\mathcal{O}$. Then, the grasp evaluator is parameterized as a neural network $\hat{y}_{\text{pen}}, \hat{y}_{\text{pick}}, \hat{y}_{\text{PGS}} = f_\varphi(\mathcal{G}, \mathcal{O})$ trained by minimizing an equally-weighted L2 loss over all 3 labels. At inference time, grasp sampling proceeds in 3 steps: (1) a large batch of candidate grasps $\mathcal{T}$ is drawn from a sampler, (2) an initial evaluation culls all but the top $K$ grasps by probability of grasp success $\hat{y}_{\text{PGS}}$ into a smaller set $\mathcal{S} \subseteq \mathcal{T}$, and (3) we iteratively refine all grasps in $\mathcal{S}$ and finally compute $\mathcal{G}^* = \arg\max_{\mathcal{G} \in \mathcal{S}} \hat{y}_{\text{PGS}}(\mathcal{G})$ afterwards.

We use a simple sampling-based update where at iterate $i$, we draw some noisy perturbation $\Delta\mathcal{G}^{(i)}$, and let $\mathcal{G}^{(i+1)} = \mathcal{G}^{(i)} + \Delta\mathcal{G}^{(i)}$ if $\hat{y}_{\text{PGS}}(\mathcal{G}^{(i)} + \Delta\mathcal{G}^{(i)}) > \hat{y}_{\text{PGS}}(\mathcal{G}^{(i)})$. Otherwise, we set $\mathcal{G}^{(i+1)} = \mathcal{G}^{(i)}$. At inference time, we do not maximize $\hat{y}_{\text{pen}}$ and $\hat{y}_{\text{pick}}$, which are only used to guide evaluator training.

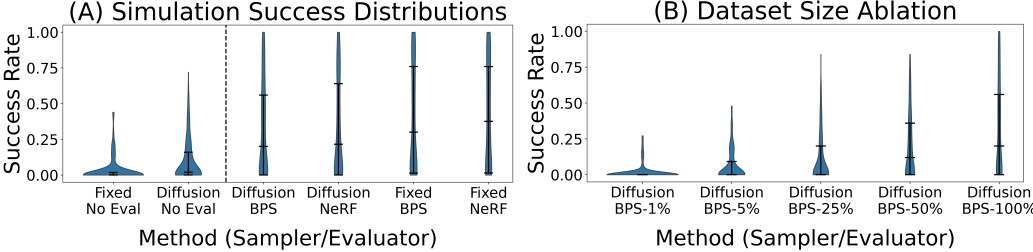

Figure 2: **Violin plots showing sim results over hundreds of unseen test objects.** The short lines mark the median and IQR. "Diffusion" vs. "Fixed" refers to sampling initial grasp $\mathcal{G}^{(0)}$ from a diffusion model or a fixed set of grasps. "NeRF" vs. "BPS" refers to whether the object $\mathcal{O}$ is given by NeRF or basis point set features. **(A)** Independent of the choice of sampler or object representation, evaluator refinement (right 4 columns) drastically improves planned grasp quality compared to sampling without refinement (left 2 columns). **(B)** Ablations on dataset size used to train the evaluator, which demonstrates a strong correlation between dataset size and grasp planner performance. The percentages indicate the fraction of the dataset used to train the evaluator.

We choose to study representations $\mathcal{O}$ that can be computed with an eye-in-hand setup. We seek to use the most favorable possible visual features to disentangle the effect of the evaluator and dataset from low-quality measurements in the real world. We move the eye-in-hand camera along a spiral-shaped trajectory that gives it diverse views of the object from many angles, yielding high-quality object reconstructions (see the right side of Fig. 10 for visualization).

Two representations compatible with the eye-in-hand are basis point sets (BPSs) [33] and NeRFs [34]. BPSs are fixed sets of points in $\mathbb{R}^3$ labeled with the closest distance to an object mesh or point cloud, essentially acting as fixed-length discretized distance fields for objects, allowing them to be inputs to networks like MLPs. NeRFs are rendering-based models that take in a set of images and their poses, producing a volumetric density field that allows novel-view synthesis. Since parallel-jaw grasping works have shown they are a promising representation, we decide to also study them for multi-finger grasping. This gives us two distinct evaluators to compare below. We sample a point cloud from the NeRF and clean it for usage in training, simulation evaluation, and hardware evaluation (see Appendix D for details on network architecture and this pre-processing).

## 4 Experiments

We hypothesize that learned grasp evaluators could help achieve robust sim-to-real transfer for multi-finger grasp planners, but only if the associated data are large-scale and account for perceptual data modalities that can be measured in the real world. We now test this hypothesis on the dataset introduced above. We also ablate other modeling choices like object representation or sampler choice to measure the performance increase uniquely attributable to the evaluator. Lastly, we measure the effect of dataset scale on evaluator performance (see Appendix A for additional analysis).

### 4.1 Simulation Results

Though our goal is robust grasping on hardware, evaluation in simulation is useful since we can test grasps on a larger set of objects. Further, we can execute multiple perturbed grasps to test grasp robustness on the identical object pose and perceptual inputs.

First, we study the importance of learned grasp evaluators by comparing with methods that do not use them. We perform an ablation study in which we vary the sampling strategy and the evaluation method. For sampling, we either use a diffusion model-based sampler (motivated by Weng et al. [25]) or a simple baseline in which we store a fixed set of ~4000 grasps from the training set (details in Appendix D.2). For evaluator refinement, we either use a BPS evaluator, a NeRF evaluator, or no evaluator, which randomly chooses grasps (can be viewed as an untrained evaluator). Figure 2A shows that all methods that use learned grasp evaluators outperform the methods that do not by a

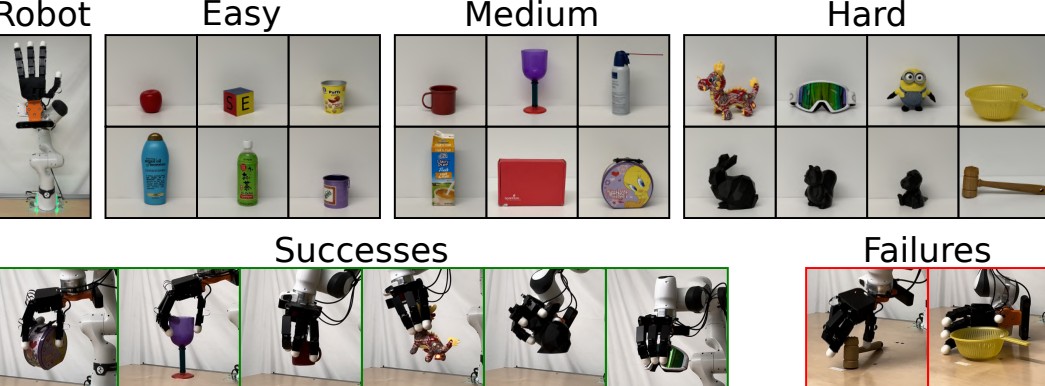

Figure 3: **Hardware setup.** Our robot consists of an Allegro hand and ZED 2i camera mounted onto a Franka Research 3. We select 20 objects categorized as easy, medium, or hard based on the complexity of their geometry, the presence of "distractor" geometry, and/or resemblance to objects in the training dataset (see App. E.2 for details). We selected these objects to maximize size/shape diversity while providing a clear upper bound on expected performance via the hard objects. The bottom row shows representative successes and failures for the "Fixed/BPS" configuration.

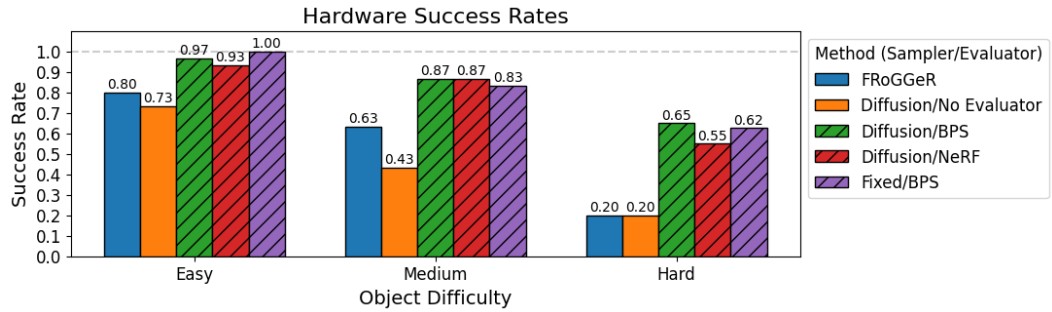

Figure 4: **Average grasping success rates on hardware across object difficulties.** We find that the methods that leverage evaluators (hatched) outperform both analytic and generative-modeling-only baselines. Detailed per-object success rates are shown in Appendix E.3.

large margin, independent of the choice of sampler or object representation. The methods that use the fixed set of grasps perform similarly with the methods that use the diffusion sampler when used with a learned evaluator, but unsurprisingly, the method that uses the fixed set of grasps performs significantly worse than the diffusion sampler.

Next, we evaluate the effect of training dataset size on evaluator performance. We perform an ablation study in which we train 4 BPS evaluator models with identical architectures and training parameters on different fractional sizes of the training dataset (1%, 5%, 25%, 50%, 100%), all of which were trained to convergence (see Fig. 19 in Appendix E). After training, we compare the performance of these models in the same simulation setup with the same fixed diffusion sampler model. Figure 2B shows that there is a strong correlation between the dataset size and grasp planner performance. It is plausible that this trend will continue with increasing dataset size, but we leave this for future work.

We evaluate all methods on a held-out set of 212 objects. We first use each method to generate 5 grasps per object. To evaluate the robustness of these grasps, we simulate each of these grasps 5 times in simulation, each perturbed with small wrist translation and rotation noise, resulting in 25 grasps per object, which is used to compute a per-object success rate.

## 4.2 Hardware Results

In this section, we evaluate whether grasp evaluators trained on our dataset improve grasping performance in the real world. We compare samplers with/without the evaluator, with/without a

diffusion-based sampler inspired by Weng et al. [25], and also vary the object representation in order to identify the factors driving performance. We also compare to FRoGGeR [22], a fast analytic method that reports strong real-world results [30] as an additional gauge for performance.

For each method, we execute 5 grasps on each of the 20 objects of varying geometry and appearance for a total of 100 grasps per method. Our dataset includes simple objects (like boxes), and adversarial ones (like reflective ski goggles), which we organize into easy, medium, and hard categories as shown in Figure 3. We remark that the baselines' reported hardware results were largely limited to convex or nearly-convex objects at low quantity [30, 25]. For a more challenging evaluation, we included more objects with non-trivial visual or geometric properties.

Our hardware setup consists of an Allegro hand mounted onto a Franka Research 3 arm with an eye-in-hand ZED 2i camera. We only use monocular RGB because the objects are too close to the camera for reliable depth readings. For repeatability, we choose a canonical pose for each object that we use for all trials. The arm follows a fixed trajectory to collect 100 NeRF training images for every grasp. For each method, we request 40 grasps from the planner (~GPU memory upper bound), then execute the highest-ranking one according to the method's metric (Diffusion No Evaluator has no metric, so the first grasp is selected). Because FRoGGeR generates grasps serially, we request only 10 grasps or as many as it can generate in 20 seconds. A grasp is considered a failure if the planner fails to return any grasps or if it does not pick the object up 20cm without dropping it. If feasible grasps are generated but the motion generation fails either during planning or execution (i.e., before the grasp), we discard the trial as the failure is not due to the grasp planner itself.

Figure 4 shows the hardware results. We find that all evaluator methods outperform the baselines by a considerable margin independent of the sampler or object representation, with the worst model in each category achieving 93%, 83%, and 55% success rates respectively. We also observe that all evaluator-based methods perform similarly, which suggests the model performance is not sensitive to the object representation. This may help disentangle representations useful for different downstream tasks from potential negative effects on grasping performance present in other learning-based methods. We discuss qualitative analysis in the Appendices A and E.3. Table 4 also shows that independent of object difficulty categorization, evaluators clearly improve performance.

## 5 Limitations

Limitations of our work are as follows. First, our hardware experiments required a time-consuming data collection trajectory and NeRF training to create good quality object models. Future work could explore training models on lower-fidelity object representations from our dataset (e.g., point cloud from 1 depth camera). Second, our methods achieve low success rates on challenging objects such as the mallet and the strainer. The performance of these models is limited by the diversity of their training data, which could be improved by training on a larger set of unique objects, including non-rigid or articulated structures. Third, our grasp generation methods are designed for tabletop settings without clutter or occluding obstacles. Future work could follow the approach used by Sundermeyer et al. [44] for parallel-jaw grasping, which trains a clutter-aware model using a no-clutter dataset.

## 6 Conclusion

We introduce a new dataset of 3.5M grasps on 4.3K unique object with RGB images, point clouds, and NeRFs integrated into our dataset, which addresses key challenges for achieving robust sim-to-real transfer of multi-finger grasping. Using this dataset, we perform comprehensive ablation studies that highlight the crucial role of learned grasp evaluators, which significantly outperform existing analytic and generative modeling-based baselines in both simulation and the real world. Our work underscores the critical importance of large-scale grasping datasets and the scale of data for required to train robust multi-finger grasp evaluators. Our contributions represent a significant step forward in the development of reliable and generalizable grasping algorithms, demonstrating the potential for large-scale data and learned evaluators to drive progress in real-world robotic manipulation.

**Acknowledgments**

We thank the reviewers for their helpful suggestions and feedback. This work is supported by the National Science Foundation under Grant Number 2342246 and by the Natural Sciences and Engineering Research Council of Canada (NSERC) under Award Number 526541680.

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

# Appendix

## A  Additional Insights and Discussion of Results

In this section, we provide further discussion that may aid the interpretation of our results.

**How "fair" is the diffusion sampler?**  To ensure reasonable performance, we train our generative model exclusively on high-quality grasps with a probability of grasp success $y_{\text{PGS}} = 1.0$. This is approximately 300K grasps (8.45% of the training data), roughly 10 times the amount used to train existing models [1, 2]. Figure 21 shows real-world grasps generated by the diffusion sampler, and shows that many failed grasps are reasonably "close" to a successful grasp.

In general, grasp evaluators have superior data efficiency in the sense that they can train on "bad" grasps (in our case, there are ~3M) which are byproducts of generating "good" ones. We stress that we do not claim that grasp evaluation is superior to grasp generation as a paradigm - only that evaluators can achieve robust sim-to-real transfer by leveraging a large volume of negative examples.

We emphasize that there are no suitable datasets for dexterous grasping larger than ours on which we can train our diffusion sampler. To our knowledge, the only large dataset with visual observations is closed-source [3]. Further, no existing datasets parameterize the grasp execution motion, few consider a tabletop setting, and many of their grasps intersect with objects in a non-physical way.

**Are the simulation results "reasonable"?**  Our simulation success rates at first appear much lower than our hardware success rates and simulation results of similar studies [1, 2]. This discrepancy arises from subtleties in labeling. In the real world, we only consider collision-free grasps, and thus report the empirically estimated conditional probability $p(\text{success} \mid \text{no collisions})$. However, in simulation we evaluate *all* planned grasps, including those in collision (which count as failures), and instead report $p(\text{success})$. In other words, the real-world experiments answer the question "how many *collision-free* generated grasps succeed?", while the simulations answer the question "how many generated grasps (collision-free or otherwise) succeed?"

Figure 5 shows histograms for all simulated evaluator-based methods, which compares unconditional success rates and those conditioned on $y_{\text{coll}} \geq 0.8$. The median success rates are 37% and 80% respectively. Note the similarity between this 80% mark and our evaluator-based hardware results, which achieve 76-81% across all objects (see Table 4). Lastly, other works differ from ours in three major ways: they often do not consider tabletop settings (allowing non-physical fully-caging grasps), they have less object diversity, and they plan power grasps while we plan precision grasps. These factors make our learning problem more challenging than those in other studies.

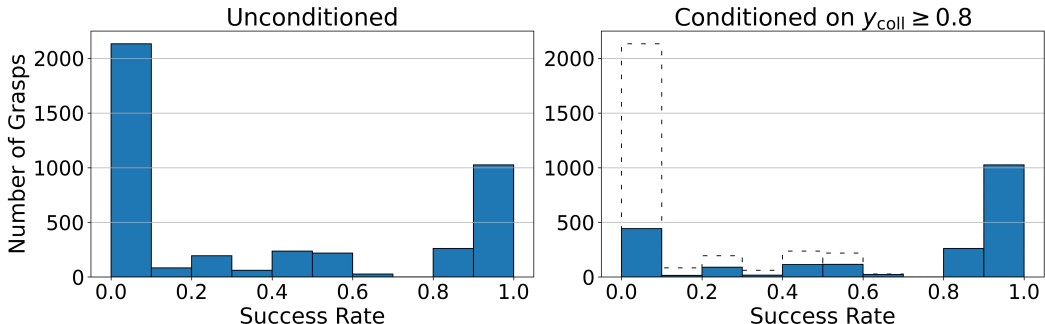

Figure 5: **Success rates for all simulated evaluator-based methods.** When only considering grasps with $y_{\text{coll}} \geq 0.8$, the median success rate increases from 37% to 80%. 2100/4240 grasps satisfied $y_{\text{coll}} \geq 0.8$. On the right, the unconditioned distribution is plotted with dashed lines for comparison.

**Interpreting the Fixed vs. Diffusion Sampler Performance.** There are a few unintuitive results regarding the fixed vs. diffusion samplers in both simulation and hardware. First, the Fixed+Evaluator methods seem to outperform the Diffusion+Evaluator methods in simulation, but they perform nearly identically in hardware. We believe that since the train and test objects in simulation are sampled from the same distribution, some test objects will resemble training objects, so some grasps from the fixed dataset may have high quality due to slight overfitting. Since there are no real-world objects in the training set, this effect disappears on hardware.

Second, it is interesting that the diffusion sampler alone far outperforms the fixed sampler, but they perform nearly identically after evaluator refinement. We believe that this is because the best samples from the fixed and diffusion samplers have similar quality, but the average diffusion-sampled grasp is far better than the average fixed grasp. Because only the best sampled grasps are further refined, and only the best of the refined grasps is executed on hardware, if the fixed sampling distribution is a reasonable prior for a good grasp, then it is likely that at least one candidate would be a good initial guess for refinement.

# B   Grasp Dataset

## B.1   Grasp Generation

| Parameter | Value | Parameter | Value | Parameter | Value |
|---|---|---|---|---|---|
| n_c_per_finger | 5 | switch_prob | 0.5 | jitter | 0.1 |
| w_fc | 0.5 | mu | 0.98 | dist_lower | 0.2 |
| w_dis | 500 | step_size | 0.005 | dist_upper | 0.3 |
| w_pen | 300.0 | stepsize_period | 50 | theta_lower | $-\pi/6$ |
| w_spen | 100.0 | starting_temp | 18 | theta_upper | $\pi/6$ |
| w_joints | 1.0 | annealing_period | 30 | | |
| w_ff | 3.0 | tempdecay | 0.95 | | |
| w_fp | 0.0 | | | | |
| w_tpen | 100.0 | | | | |

Table 3: **Grasp generation parameters.** Variables names taken from DexGraspNet [4]. Left: Energy function parameters. Middle: Optimization parameters. Right: Initialization parameters.

Our grasp generation pipeline is inspired by Wang et al. [4], but required key modifications. Further details about these modifications not discussed in the main text are described here.

- As mentioned, our data generation pipeline yields *pre-grasp* poses with the fingertips 1.5cm off of the surface of the object, while DexGraspNet places the fingers on or very near the surface. Due to this, when planning a grasp trajectory, we compute a new configuration corresponding to fingertip locations 3.5cm below the object surface, which we call the *post-grasp* pose.

- We choose to generate *precision grasps* rather than power grasps, since this synergizes with the grasp motion parameterization described in the main text. To implement this, we only specify *contact candidates* on the fingertip of the hand, which function as attractors to the object surface during grasp generation. However, the *surface points* used to penalize hand-object or hand-table collision still cover the whole hand. See Figure 6 for a visualization.

- We sample 500 surface points uniformly from the mesh of each link's collision geometry, distributed proportionally based on mesh surface area. 128 contact candidates are densely sampled over the "front" of each Allegro fingertip, which is parameterized by a spherical section centered on the (spherical) fingertip center. This spherical section has a radius of 1.2cm, a polar angular sweep of $\pm 54°$, and an azimuthal angular sweep of $54°$ to $126°$. The local fingertip coordinate frame has the $+x$ axis pointing in the "palm out" direction and the $+z$ axis aligned with the fingers.

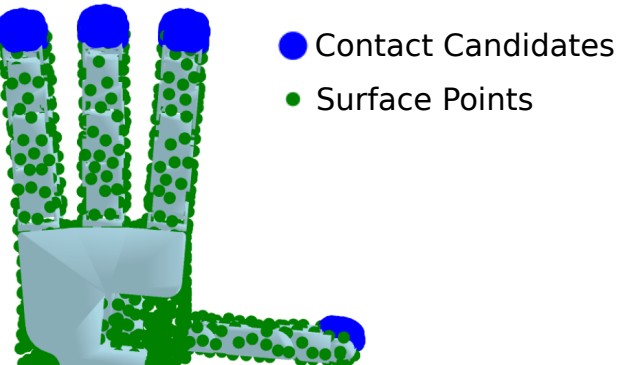

Figure 6: **Visualization of the Allegro hand contact candidates and 500 surface points.** Contact candidates are used to compute $E_{\text{dis}}$, encouraging hand-object proximity. Surface points are used to compute $E_{\text{tpen}}$, which penalizes hand-table penetration to keep the grasps above the table.

We now describe the dataset augmentation procedure in detail. Recall that first, a large set of *nominal* grasps are generated using the modified DexGraspNet pipeline. In this step, the fingertip grasp directions $\mathbf{d}_i$ are generated by computing the direction of each fingertip to the closest point on the mesh. Because there are a large proportion of failures after this step, we augment the dataset with more positive examples by taking all nominal grasps satisfying $y_{\text{PGS}} \geq 0.9$, perturbing them slightly, re-evaluating them, and adding them to the dataset.

These perturbations are sampled using Halton sequence sampling [5], which generates quasi-random, *low-discrepancy* sequences that sample more uniformly across the input space compared to standard random sampling and reducing clustering. We sample perturbations from the following ranges:

- wrist translation: each spatial coordinate draws a perturbation from [-5mm, 5mm];
- wrist orientation: each of roll, pitch, and yaw draw a perturbation from [-2.5°, 2.5°];
- finger joints: each angle draws a perturbation from [-0.05rad, 0.05rad];
- fingertip grasp directions: each of two axes orthogonal to the nominal direction draw perturbations from [-10°, 10°].

Each high-success grasp is perturbed five times and re-evaluated on the same object. Additionally, these grasps are perturbed two more times and evaluated on *different, randomly sampled* objects from the dataset. These additional perturbations typically yield unsuccessful grasps, but we believe this protects the model from overfitting and provides useful training signal regarding object geometry. Table 3 summarizes all parameters used for grasp generation.

## B.2  Grasp Label Generation in Simulation

To generate the labels for the grasps in the dataset as well as corresponding rendered visual data, we use Isaac Gym [6]. Our simulated evaluation proceeds as follows. We spawn the hand in the pre-grasp pose, execute the grasp motion parameterized by fingertip directions, then lift the object 20cm off of the table (with gravity enabled).

A pick is considered a *simulation success* ($y_{\text{pick}} = 1$) only if the following conditions are met: (1) the hand contacts the object on at least 3 links; (2) the palm and proximal links of the hand do not touch the object; (3) the change in the object's position relative to the hand does not exceed 10cm throughout the lift; (4) the change in the object's orientation relative to the hand does not exceed 45 degrees about each Euler angle throughout the lift. A pick is considered *collision-free* ($y_{\text{coll}} = 1$) only if the hand does not contact the table or object during the pre-grasp pose. Recall that $y_{\text{PGS}} = y_{\text{pick}} \wedge y_{\text{coll}}$.

# Simulated Grasp Evaluation

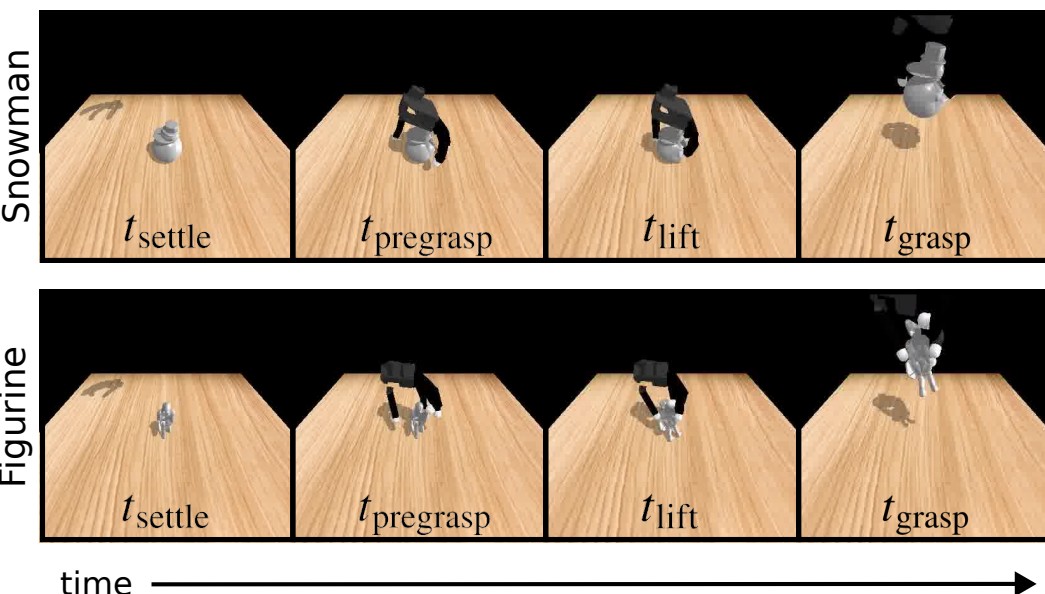

Figure 7: **Visualization of grasp evaluation in simulation on a snowman object (top) and a figurine object (bottom).** After the object settles on the table ($t_{\text{settle}}$), the Allegro hand is moved to the pre-grasp position ($t_{\text{pregrasp}}$), executes the grasp ($t_{\text{grasp}}$), and lifts the object ($t_{\text{lift}}$). At $t_{\text{pregrasp}}$, we compute $y_{\text{coll}}$ by checking for hand collisions with the object and table. At $t_{\text{lift}}$, we compute $y_{\text{pick}}$ by checking if the object pose relative to the hand significantly changed from $t_{\text{pregrasp}}$.

To generate non-binary labels, each grasp is corrupted with small noise and re-simulated 5 times. Note that this is a *different perturbation* than the one used for dataset augmention described in Appendix B.1. Only the wrist pose is perturbed. The wrist translations are corrupted with uniform noise drawn from [-5mm, 5mm] along each axis, and the wrist orientation is corrupted about the roll, pitch, and yaw axes with uniform noise drawn from [-2.5°, 2.5°].

The simulation timestep is set to 1/60 seconds, but the integrator solves at a rate of 1/120 seconds for numerical stability using the Truncated Gauss-Seidel (TGS) solver. The TGS solver runs 8 position and 8 velocity iterations per simulation timestep. "Force at a distance" (i.e., the *contact offset* parameter) is applied starting from 1mm of separation between collision geometries, substantially lower than the 1cm distance used by [4] in DexGraspNet. This helps reduce unwanted non-physical collisions. Geometries are processed by a convex decomposition using the default settings in Isaac Gym. Each object is spawned 2cm above the table and dropped. The simulation starts when the object settles. The simulation pipeline is visualized in Figure 7.

## B.3 Simulated Label Distribution

Figure 8 shows the distribution of grasp labels across the dataset of 3.5M grasps. The median and IQR of each label is as follows:

- $y_{\text{PGS}}$: 0.0 (0.0, 0.48)
- $y_{\text{pick}}$: 0.2 (0.0, 0.8)
- $y_{\text{coll}}$: 0.08 (0.6, 1.0)

## B.4 Object Dataset and Generation

The objects considered in this work are a strict subset of those in DexGraspNet [4]. DexGraspNet contains 5.3K unique objects, while we only consider 4.3K of them. The main reason for this is

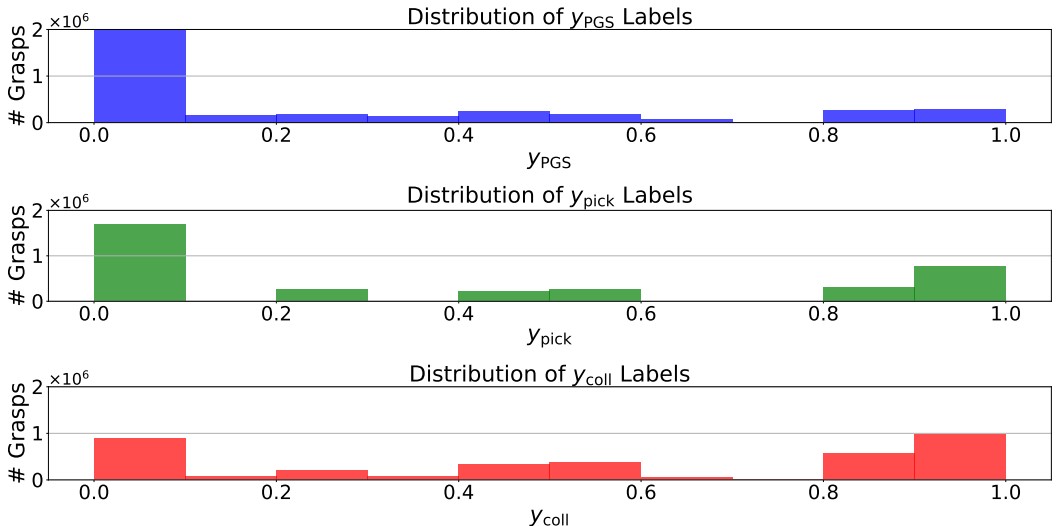

Figure 8: **Dataset label distribution across 3.5M grasps.**

that we assume a tabletop setting while DexGraspNet does not, which necessitates a canonical "up" direction. Thus, many objects were manually processed to be oriented in a reasonable manner. We only use an object if after this reorientation, the object successfully settles when dropped from a height of 2cm. If the object tips over or does not settle after 200 timesteps in the Isaac Gym simulation, then it is excluded. We consider an object to have remained upright if the real part of its relative quaternion remains greater than 0.95 throughout the simulation. We consider it to have settled if the translational components stay within 1mm along each axis and roll, pitch, and yaw stay with 1e-2 radians for 15 consecutive timesteps. For all objects, we used a uniform friction coefficient of 0.9 and a constant object density of 500 kg/m$^3$ (as in [4]). Thus, object masses were determined entirely by volume, as in other works [7, 4, 8].

Figure 9 visualizes a subset of our object dataset. To introduce more scale diversity while retaining a dataset size under 2.5TB, a subset of approximately 1.7K of the objects is duplicated at three distinct scales, while the remaining objects are used at one scale each, resulting in approximately 7.7K objects. The large dataset storage requirement comes from the storage of multiple 3D NeRF density grids used to represent each grasp for the NeRF evaluator (more details about the NeRF density grid representation in Appendix D.3.2).

### B.5 Train, Validation, and Test Split

We create our train, validation, and test splits of the datasets very carefully. We make sure that each unique object mesh only exists in one of these three sets.

We split the 4305 unique object meshes into 3981 (92.5%) train objects, 216 (5%) validation objects, and 108 (2.5%) test objects. Each object may appear at more than one scale, so this results in 7695 different objects after all scaling operations, split into 7108 (92.5%) train objects, 375 (5%) validation objects, and 212 (2.5%) test objects. This results in a total of 3,531,098 grasps, with 3,261,228 (92.5%) for training, 170,128 (5%) for validation, and 99,742 (2.5%) for testing.

### B.6 Compute and Timing

We used 4 Nvidia A100 GPUs to generate our dataset. It took ~4 days to generate the full dataset: ~1 day for grasp generation, ~1 day for grasp label generation, and ~2 days for image generation, NeRF training, and point cloud generation.

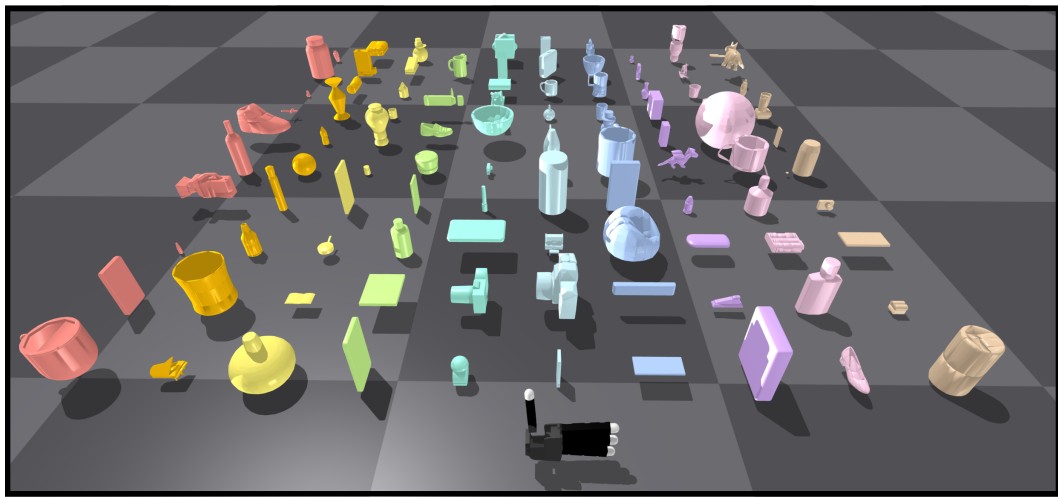

Figure 9: **Visualization of a random subset of 100 objects out of the 212 test objects.** The Allegro hand is shown at the bottom for scale. The object dataset is a subset of the DexGraspNet dataset's meshes [4]. See their work for more details.

## C  Object Representations

### C.1  Neural Radiance Fields

We train NeRFs using both simulated data and real-world data and compare them here.

For NeRFs trained on simulated data, the object is first spawned and allowed to settle on the table. Then, we uniformly sample 100 images over a spherical cap above the object with a radius of 0.45m and a polar angle sampled from $U(0, \frac{\pi}{4})$, and then train for 400 iterations with nerfstudio [9, 10].

For NeRFs trained on real-world data, we first collect 100 images while moving the wrist-mounted camera along a hard-coded trajectory encircling the object and then train for 400 iterations with nerfstudio [9, 10]. This trajectory consists of 3 spirals around the object, with the object placed roughly in the center of the spirals.

Figure 10 shows a qualitative comparison between the RGB images and NeRFs trained on simulated data versus real-world data. Both types of NeRFs exhibit floater artifacts [11], which must either be processed away or ignored by downstream models.

We train without scene contraction, auto-scaling poses, centering method, or orientation method, and use a scale factor of 1.0, which ensures that the coordinate space in the NeRF is not modified from the given data. The remainder of the parameters follow the default nerfacto settings in nerfstudio.

### C.2  Basis Point Sets

Methods that represent the object as basis point sets (BPSs) use the following procedure.

1. Train a NeRF following the procedure described in Appendix C.1.

2. Sample a point cloud with 5000 points using nerfstudio [10] by rendering out depth images with opacity exceeding 0.5 in the axis-aligned bounding box parameterized by the lower and upper bounds $[-0.2m, -0.2m, 0.0m] \times [0.2m, 0.2m, 0.3m]$ using a z-up convention.

3. Preprocess the point cloud to remove most outliers and floaters. First, we use open3d [12] to remove statistical outliers (nb_neighbors=20, std_ratio=2.0) and radius outliers (nb_points=16, radius=0.05). Next, we construct an undirected graph of the

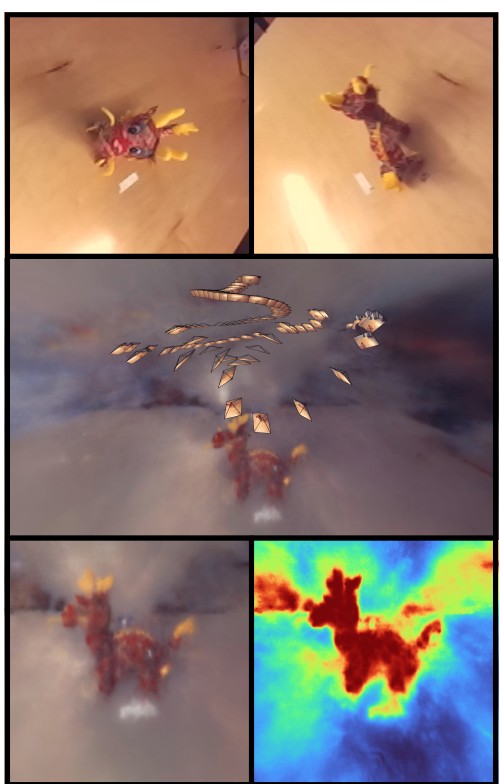

Figure 10: **Comparison between NeRFs trained on simulated snowman data and real-world dragon data.** This demonstrates the qualitative similarity between the two NeRFs. First row: two of the RGB images used for NeRF training. Second row: Camera poses used for NeRF training. Third row: NeRF-rendered RGB image and accumulation image.

      remaining points, where points are nodes and an edge exists between two nodes if the points are within 1mm. Floaters are removed by keeping the largest connected component.

4. Generate a random set of 4096 basis points within a sphere of radius 0.3m, centered 0.15m above the table. This set of basis points remains fixed for all experiments. We utilize `bps` [13] for basis point set operations.

5. Compute basis point set values by calculating the distance from each basis point to its closest point in the point cloud.

Figure 11 shows an example of a point cloud and BPS for both a simulated and real-world object.

### C.3   Meshes

We use triangle meshes for two purposes: the analytic grasp planning method (FRoGGeR) that we use as a baseline and collision-free motion planning from the start pose to the pre-grasp pose (all methods).

For both applications, the mesh is generated by the following procedure.

1. Training a NeRF following the procedure described in Appendix C.1.

2. Using `scikit-image` [14] to perform marching cubes, which extracts a 2D surface mesh from a 3D volume with a NeRF density level set of 15 within a bounding box of $[-0.2m, -0.2m, 0.0m] \times [0.2m, 0.2m, 0.3m]$ (with z being the up-direction).

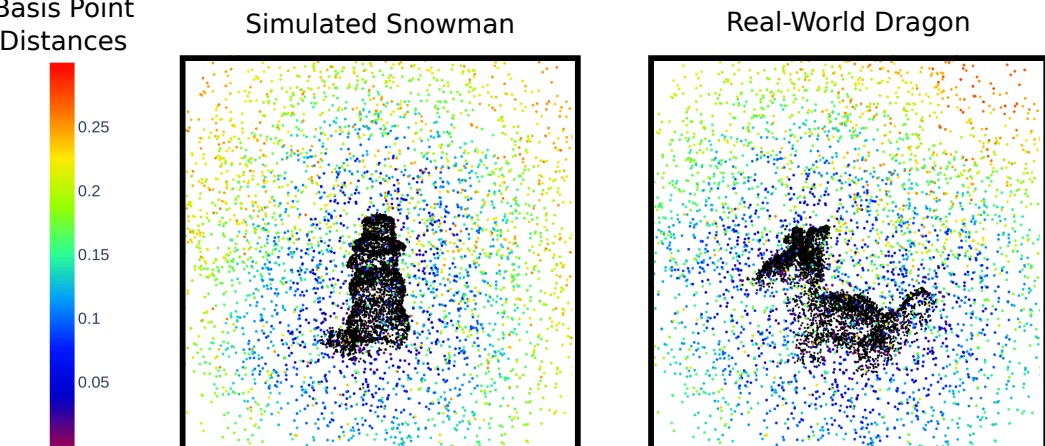

Figure 11: **Visualization of a point cloud (5000 black points) and a basis point set (4096 points colored by distance to the point cloud).** These are generated by a NeRF for both a simulated snowman object (left) and a real-world dragon object (right).

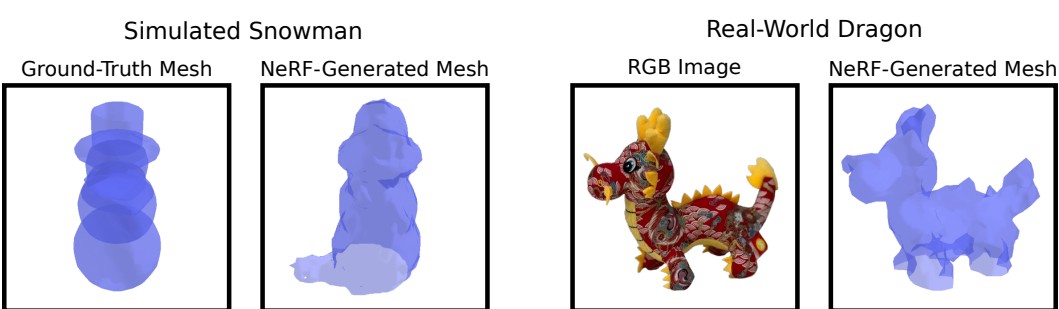

Figure 12: **We use NeRFs to generate meshes for two purposes: analytic grasp planning methods (FRoGGeR) and collision-free motion planning from the start pose to the pre-grasp pose (all methods).** Left: Comparison between the ground-truth mesh and NeRF-generated mesh of a simulated snowman object, where the extra vertices at the bottom are due to artifacts from shadows/lighting effects. Right: Comparison between an RGB image and the NeRF-generated mesh of a real-world dragon object. We do not have a ground-truth mesh for the real-world dragon object.

3. Using `trimesh` [15] to remove floaters. This is done by only keeping connected components with at least 31 edges.

Figure 12 shows examples of NeRF-generated meshes. We found the parameters above to be reasonable for all test objects.

## D   Grasp Planning

As explained in the main text, during grasp planning, a batch of candidate grasps is sampled, and only the top $K$ of these are retained for the refinement phase. In our simulation experiments, we let $K = 5$, which are all refined and executed in simulation. In our real-world experiments, we let $K = 40$, which are all refined and the best is executed on hardware. See Figure 13 for a schematic of the sample/refine process.

Recall that our refinement is sampling-based, where the previous iterate is perturbed by some number of samples and the best one is chosen to be the next iterate. For all perturbations, we use zero-mean Gaussian noise as follows.

- Wrist translation: standard deviation of 5mm per spatial coordinate.

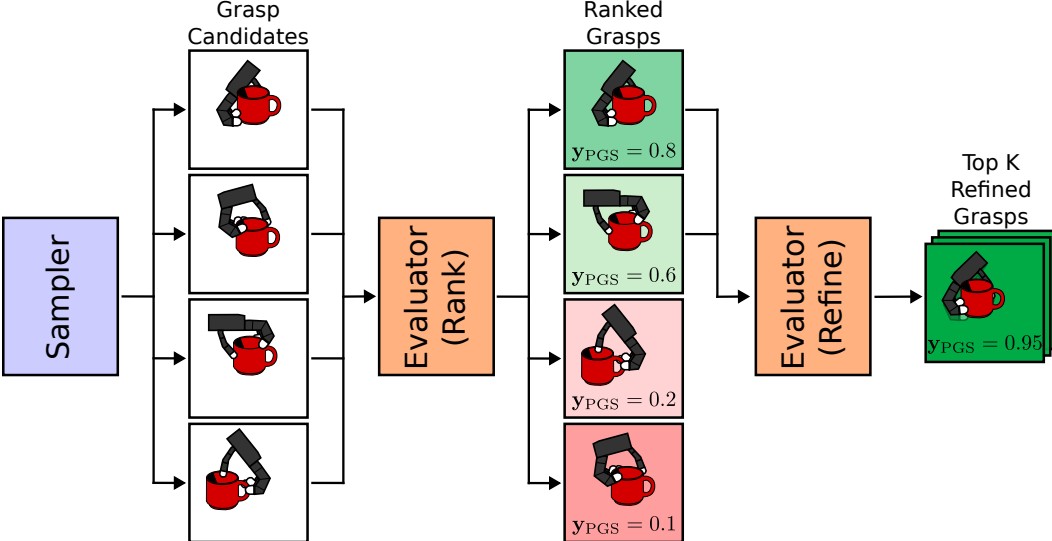

Figure 13: **Our grasp planners consist of a *sampler* and an *evaluator*.** First, the sampler generates a batch of grasp candidates. The evaluator ranks them, and the top $K$ grasps are refined using sampling-based optimization, where the objective is given by the evaluator.

- Wrist and grasp orientations: the noises are sampled using a standard deviation of 0.05m from $\mathfrak{so}(3)$, which are then converted to elements of $SO(3)$ via the exponential map.
- Joint angles: standard deviation of 0.01 radians.

The refinement is run for 50 iterations. Other refinement strategies could work as well, such as gradient-based methods or other sampling-based methods.

To allow grasp parameters to be represented as vector inputs or outputs, we parameterize them as $\mathbf{g} = (\mathbf{x}, \mathbf{r}, \theta, \mathbf{d}_1, ..., \mathbf{d}_{n_f}) \in \mathbb{R}^{9+n_j+3n_f}$, where $\mathbf{x} \in \mathbb{R}^3$ is the wrist position, $\mathbf{r} \in \mathbb{R}^6$ is the wrist orientation represented by a continuous 6-D rotation vector [16], $\theta \in \mathbb{R}^{n_j}$ is the pre-grasp joint configuration, and $\mathbf{d}_i \in \mathbb{R}^3$ is the direction in which the $i^{\text{th}}$ fingertip moves during the grasp.

### D.1 Diffusion Sampler

Figure 14 shows the diffusion sampler architecture inspired by Weng et al. [1]. As their implementation is not publicly available, we re-implemented their architecture to the best of our ability via the textual description. We use an embedding dimension of 128 and a sequence length of 4 for all attention modules. The sequence length is created by reshaping the outputs of the fully-connected layers (to (4, 128)). We use Denoising Diffusion Implicit Models (DDIM) [17] with a linear scheduler on the noise variance $\beta$ that moves from 0.0001 to 0.02 over a total of 1000 diffusion timesteps. Figure 15 shows the train and validation loss curves for this model.

Although the diffusion sampler is trained to only generate successful grasps, it can still produce unsuccessful grasps as seen in previous works [1, 2]. While it could typically generate feasible-looking grasps on a diverse range of objects, its most common failure modes were (1) generating grasps that were "close" to successful but with fingers slightly too close or far to successfully execute, and (2) generating a grasp that was substantially too far from the object.

### D.2 Sampling from a Fixed Dataset

To disentangle the importance of the learned evaluator from the impact of the diffusion sampler, we perform additional experiments in which we replace the diffusion sampler with a simple baseline in which we store a fixed set of 4349 grasps from the training set. More specifically, this was

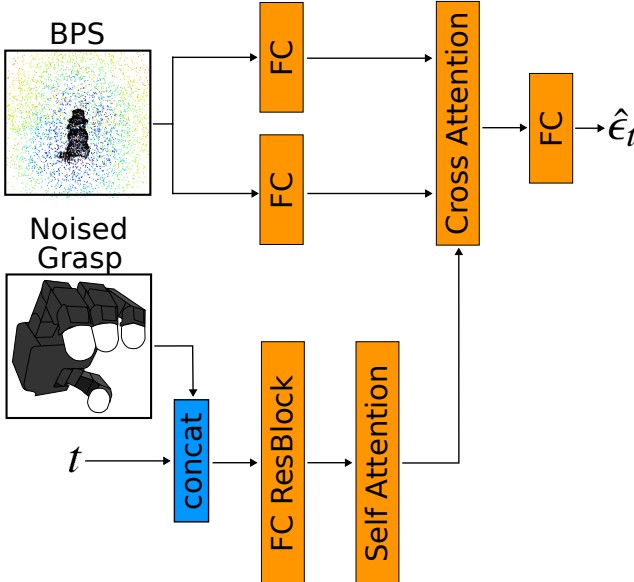

Figure 14: **Diffusion Sampler architecture.** Our model takes in a basis point set, a noised grasp, and a diffusion timestep. The noised grasp and the diffusion timestep are processed into a query using a self-attention block. The basis point set is processed into a key-value pair. These are embedded together using a cross-attention block and used to compute the predicted noise. This is a similar architecture to the one used by Weng et al. [1].

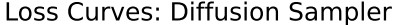

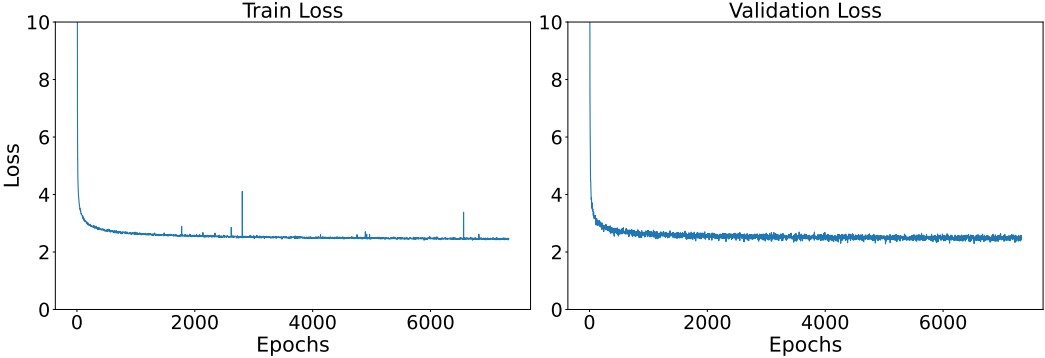

Figure 15: **Train and validation loss curves for the diffusion sampler.**

generated by storing one grasp per training set object that exhibited a high probability of success ($y_{\text{PGS}} \geq 0.9$). For objects that did not have any such grasps, we did not store any. We emphasize that these grasps were from training set objects, so they were not designed for the unseen test objects used for simulation evaluation or the real-world objects used for hardware evaluation.

## D.3 Grasp Evaluators

Recall that the evaluators are trained by regressing on three distinct soft labels: $y_{\text{PGS}}$, $y_{\text{coll}}$, and $y_{\text{pick}}$. Although we only use the $\hat{y}_{\text{PGS}}$ prediction at inference time, we choose to regress the other two labels $\hat{y}_{\text{coll}}$ and $\hat{y}_{\text{pick}}$ at train time, as we found that these labels provide an additional signal about *why* a given grasp may be failing.

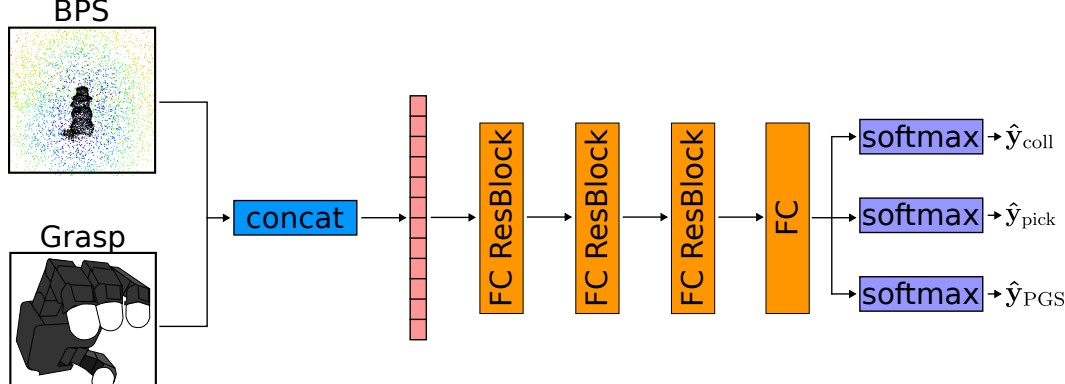

Figure 16: **BPS Evaluator architecture.** Our model takes in a basis point set and a grasp, which are concatenated together and then passed through three fully-connected resblocks and a final fully-connected layer, which returns logits for 3 labels: whether (i) there are unwanted collisions in the scene, (ii) the pick succeeded in the simulator, and (iii) both (i) and (ii) are true. See the official implementation from Mayer et al. [2] [here]. This is the same architecture used by Weng et al. [1] and our work.

### D.3.1 BPS Evaluator Details

Figure 16 shows the architecture used for the BPS evaluator. We used the official implementation from Mayer et al. [2] [here], which is the same architecture used by Weng et al. [1].

### D.3.2 NeRF Evaluator Details

Figure 17 shows the architecture used for the NeRF evaluator. We use a novel grasp representation that leverages NeRF features directly.

We will motivate this representation and then describe it in detail. A key factor for grasp success is the surface geometry at the contact points. Modeling this from images is difficult, and few (if any) fast, reliable surface reconstruction methods are adequate for grasp planning. Here, we use NeRF features that we hypothesize capture accurate estimates of the object surface. Centered at finger $i$'s position $\mathbf{x}_i$, a square of side length 0.06m is swept along $\mathbf{d}_i$ by 0.08m to recover a rectangular prism which is discretized into a 4D tensor $\mathbf{N}_i \in \mathbb{R}^{4 \times 31 \times 31 \times 41}$, where the first channel dimension is the NeRF density and the last 3 are spatial coordinates. These grid dimensions overapproximate both the fingertip size and the grasp depth to ensure the geometric information captured is not too local. Further, to capture the global object geometry, we generate a grid centered on the estimated object centroid with side lengths 0.4m $\mathbf{N}_g \in \mathbb{R}^{4 \times 41 \times 41 \times 41}$. The centroid is estimated by assuming uniform mass density and integrating over spatial regions with NeRF density exceeding 15.0.

In summary, our architecture uses local NeRF densities and global NeRF densities as inputs. The local NeRF densities are sampled grids approximating the fingertip swept volumes when moving along the grasp directions. The global NeRF densities are a sampled grid of fixed size to capture the object's global geometric features.

### D.4 FRoGGeR Details

For all experiments that use FRoGGeR [18, 19], we use the open-source implementation provided at `https://github.com/alberthli/frogger`. The procedure for acquiring the meshes used for planning is described in Appendix C.3.

Out of the 100 real-world pick attempts using FRoGGeR, there were a total of 49 failures. 16 failures were caused by planned grasps that failed to lift the object, and 33 failures were caused by planning issues in which no grasp could be found within the timeout period.

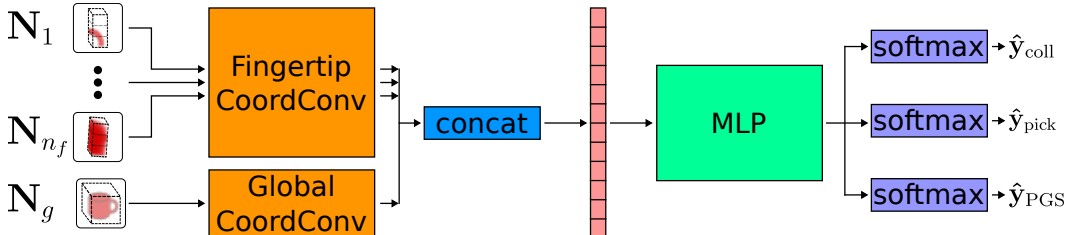

Figure 17: **NeRF Evaluator architecture.** Our model takes in local and global NeRF feature grids, passing them through 3D CoordConv networks and an MLP, which returns logits for 3 labels: whether (i) there are unwanted collisions in the scene, (ii) the pick succeeded in the simulator, and (iii) both (i) and (ii) are true. The Fingertip CoordConv weights are shared for all fingertips.

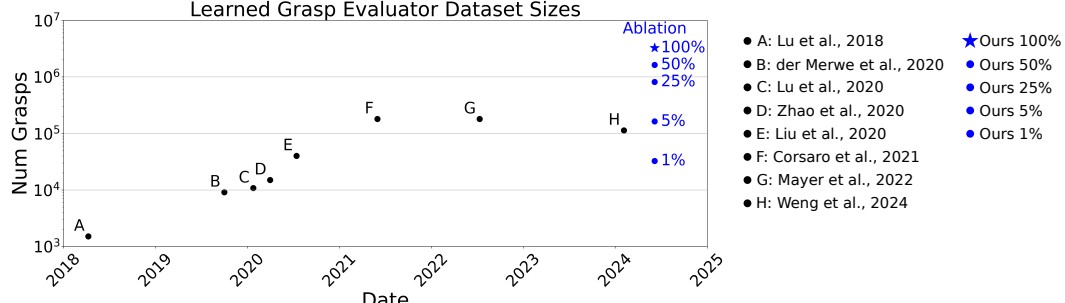

Figure 18: **Comparison between learned grasp evaluator dataset sizes in prior works over time.** Our dataset and our ablations shown on the right in blue. The full training set contains 3.26M grasps.

FRoGGeR allows users to either plan with a floating hand or the full hand-arm system. We opt to use the full hand-arm system so that the generated grasps account for kinematic reachability of grasp poses when using a given arm, resulting in fewer downstream motion planning failures.

# E    Experiment Details

## E.1    Simulation: Ablation Study on Dataset Size Details

We hypothesize that learned grasp evaluators could help achieve robust sim-to-real transfer for multi-finger grasp planners, but only if the associated data are large-scale and account for realistic perceptual inputs.

Looking at prior works, we see a clear trend toward larger multi-fingered grasp datasets used for training grasp evaluators (see Figure 18). Recognizing this trend, our goal with this ablation is to study how much the increased scale of our dataset (3.5M grasps across 4.3K objects) impacts performance, which we measure by examining the improvement in the simulated probability of grasp success gained by evaluator refinement.

Figure 19 shows the train and validation loss curves of each model trained on a different fraction of the training dataset and validated on the same validation dataset. We see that all models are trained to convergence and achieve similar train losses, but their corresponding validation losses show a consistent correlation between dataset scale and validation performance.

## E.2    Hardware: Object Selection

Figure 20 shows the real-world objects labeled with their corresponding names. Easy objects are those that have high-quality grasps when approached from any or most directions. For example, tall, cylindrical objects can be grasped overhead or from the side. Medium objects are those with more

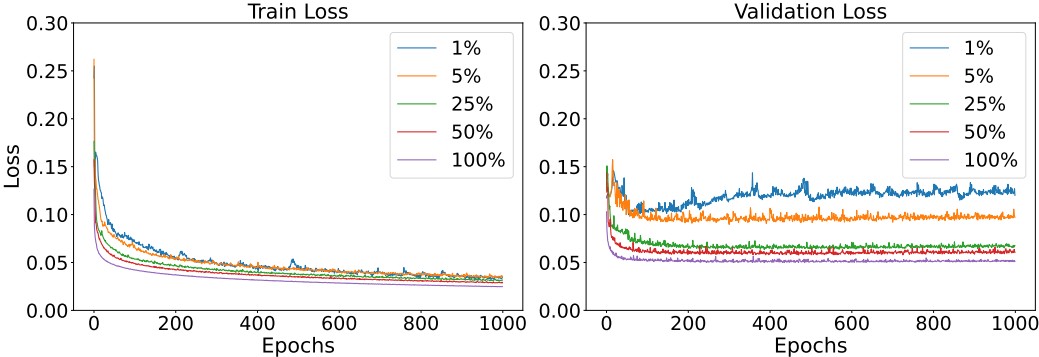

Figure 19: **Comparison of loss curves across each model trained on a different fraction of the training dataset and validated on the same validation dataset.** These show a clear correlation between dataset scale and validation performance. The full training set contains 3.26M grasps.

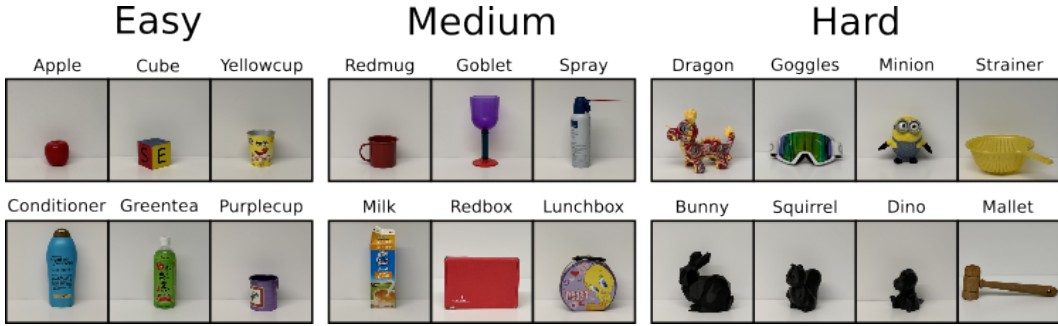

Figure 20: **Real-world objects labeled with their names corresponding to Table 4.**

unusual geometry, including "distractors," or objects which are harder to grasp when the hand is arbitrarily oriented. For example, the mug has a handle and the lunchbox is more easily pinched in the thin direction. Lastly, hard objects contain very unusual, non-convex geometry/visual features, or are difficult to grasp without affordances. For example, the goggles are both reflective and transparent, which poses a challenge for object reconstruction algorithms, and the bunny's ears are a large distractor for grasp planners. The bunny, squirrel, and dino were custom printed as hard test objects. The strainer and mallet were chosen to be out-of-distribution objects, providing a ceiling on the performance of our grasping algorithms. No method obtained any successful grasps on the strainer.

### E.3 Hardware: Detailed Results

Table 4 presents the detailed quantitative results of all grasp planning methods on various objects in the real world.

Qualitatively, we observed that our evaluator-based methods very rarely exhibited *edge-seeking* behavior, wherein the fingers are placed on corners or edges of objects, and has been noted as a common failure mode by other works [18, 19]. We suspect this may be a result of our label smoothing, though we do not rigorously test this. However, "distractor" geometries like the bunny ears or spray nozzle frequently caused grasp planning failures, attracting the fingers towards them. Methods like FRoGGeR rely on online mesh reconstruction, so objects like the goggles caused issues due to transparency/reflectivity. Lastly, many of the diffusion sampler's unrefined grasps appeared "close" to success even when failing, which suggests that generative modeling may be a viable strategy with more data. Some examples are shown in Figure 21.

| Difficulty | Objects | Methods | | | | |
|---|---|---|---|---|---|---|
| | | No Evaluator | | Evaluator | | |
| | | FRoGGeR | Diffusion No Evaluator | Diffusion BPS | Diffusion NeRF | Fixed BPS |
| Easy | Apple | 4/5 | 3/5 | 5/5 | 5/5 | 5/5 |
| | Cube | 3/5 | 1/5 | 5/5 | 4/5 | 5/5 |
| | Yellowcup | 5/5 | 4/5 | 5/5 | 5/5 | 5/5 |
| | Conditioner | 4/5 | 5/5 | 5/5 | 4/5 | 5/5 |
| | Greentea | 3/5 | 4/5 | 4/5 | 5/5 | 5/5 |
| | Purplecup | 5/5 | 5/5 | 5/5 | 5/5 | 5/5 |
| | TOTAL | 24/30 (80%) | 22/30 (73%) | 29/30 (97%) | 28/30 (93%) | **30/30 (100%)** |
| Medium | Redmug | 2/5 | 4/5 | 5/5 | 5/5 | 5/5 |
| | Goblet | 5/5 | 3/5 | 5/5 | 5/5 | 5/5 |
| | Spray | 4/5 | 0/5 | 2/5 | 4/5 | 3/5 |
| | Milk | 1/5 | 5/5 | 5/5 | 5/5 | 5/5 |
| | Redbox | 5/5 | 1/5 | 4/5 | 5/5 | 3/5 |
| | Lunchbox | 2/5 | 0/5 | 5/5 | 2/5 | 4/5 |
| | TOTAL | 19/30 (63%) | 13/30 (43%) | **26/30 (87%)** | **26/30 (87%)** | 25/30 (83%) |
| Hard | Dragon | 0/5 | 0/5 | 2/5 | 2/5 | 4/5 |
| | Goggles | 0/5 | 0/5 | 4/5 | 2/5 | 4/5 |
| | Minion | 0/5 | 3/5 | 5/5 | 5/5 | 5/5 |
| | Strainer | 0/5 | 0/5 | 0/5 | 0/5 | 0/5 |
| | Bunny | 0/5 | 1/5 | 5/5 | 4/5 | 3/5 |
| | Squirrel | 1/5 | 2/5 | 4/5 | 3/5 | 4/5 |
| | Dino | 2/5 | 2/5 | 5/5 | 5/5 | 5/5 |
| | Mallet | 5/5 | 0/5 | 1/5 | 1/5 | 0/5 |
| | TOTAL | 8/40 (20%) | 8/40 (20%) | **26/40 (65%)** | 22/40 (55%) | 25/40 (62%) |
| All | TOTAL | 51/100 (51%) | 43/100 (43%) | **81/100 (81%)** | 76/100 (76%) | 80/100 (80%) |

Table 4: **Results of different grasp planning methods on various objects in the real world.** Images of these objects can be found in Figure 20.

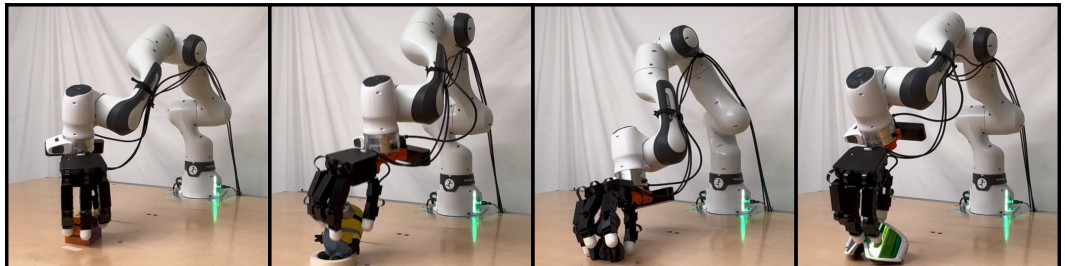

Figure 21: **Some examples of failed grasps generated by the diffusion sampler.** Though they failed, the grasps appear qualitatively reasonable.

### E.4 Hardware: Motion Planning

To evaluate a given grasp in the real world, we need to use a motion planner to move the hand to the corresponding pre-grasp pose. We use cuRobo [20], which performs parallelized collision-free motion generation. In particular, we use cuRobo's graph planner, which uses Probabilistic Road Map (PRM) to find a collision-free path. Object collision avoidance requires an object mesh, which is acquired through a process explained in Appendix C.3. We use a collision buffer of 1mm.

When executing a grasp on hardware, we decompose the motion into three stages: (1) start pose to pre-grasp pose, (2) pre-grasp pose to grasp pose, (3) grasp pose to lift pose.

The first stage is the most challenging motion planning problem because the hand needs to find a collision-free path to get very close to the object. To simplify this problem, we utilize inverse kinematics to adjust the pre-grasp finger joints, moving them an additional 3cm backward along the fingertip grasp directions.

In addition to ensuring kinematic feasibility, we also need to avoid damaging the cabling of the wrist-mounted camera during grasp execution. To achieve this, we filter out grasp samples if either (1) the normal direction of the palm is within 60 degrees from the upward direction of the world frame or (2) the direction from the palm to the middle finger deviates by more than 60 degrees from the forward direction of the world frame.

To account for these checks, we request 40 grasps from the grasp planner, which are then sorted by the grasp planner's metric. If no metric is available, such as in the case of a diffusion sampler without an evaluator, the grasps remain unsorted. Next, we utilize `cuRobo` to solve all 40 motion planning problems in parallel and then execute the first grasp for which a motion plan is successfully found.

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
