# OpenReview forum: "Get a Grip: Multi-Finger Grasp Evaluation at Scale Enables Robust Sim-to-Real Transfer"
_robot-learning.org/CoRL/2024/Conference — CoRL 2024_

### Official Review · Reviewer_k59v · 2024-07-17
**Very useful dataset contribution, unsure about how the dataset connects to grasp evaluator hypothesis and subsequent experiments ran**

**Originality:** 3
**Technical Quality:** 3
**Clarity Of Presentation:** 4
**Potential Impact:** 3
**Recommendation:** 2
**Confidence:** 5

**Review:**

1. Missing references (non exhaustive list):

When Transformer Meets Robotic Grasping: Exploits Context for Efficient Grasp Detection. S. Wang, Z. Zhou, and Z. Kan.

Geometry Matching for Multi-Embodiment Grasping. M. Attarian, M. A. Asif, J. Liu, R. Hari, A. Garg, I. Gilitschenski, J. Tompson

Grasp Proposal Networks: An End-to-End Solution for Visual Learning of Robotic Grasps. C. Wu, J. Chen, Q. Cao, J. Zhang, Y. Tai, L. Sun, K. Jia

2. Ln. 56-57: “condition on images taken from a trajectory encircling the object”, re: unclear sentence. What does this mean exactly?

3. Ln. 59-60: “We find that dataset scale is necessary for training effective grasp evaluators, which significantly degrade in performance when trained on smaller subsets of the data.”. I would expect this to also hold for generative methods without an evaluator.

4. Ln. 70-71: “We test our evaluators trained on our dataset on thousands of simulated and hundreds of real-world grasps, and demonstrate state-of-the-art performance.”. Surprised to see a SoTA claim. Is there a new evaluator method introduced in this paper? Impression left from the introduction was that multiple existing evaluators are used. If there is a new evaluator method introduced that achieves SoTA, please introduce in introduction section.

5. Ln. 130-134: It’s not completely obvious or intuitive that perturbing grasps that are more likely to pass, and reevaluating, yields more grasps that are likelier to pass.

6. Ln. 154: Unclear why directions of fingers are needed for ambiguity of grasp execution. Typically each joint has an angle range that it can take that corresponds to a configuration and the finger will move in whatever direction it needs to to get to that configuration.

7. Fig. 3: Sounds like the categorization of objects in easy, medium, hard is subjective. Complexity of geometry: how is that defined/measured? Resemblance with objects in the training set: how is that measured?

8. Only one baseline included for hardware results. I’d argue that diffusion/no evaluator is not a baseline, it’s an ablation.

9. The claim that evaluators are strictly outperforming analytic and generative methods without an evaluator, is a bit of an overclaim. There was no comparison provided to any generative methods.

**Quality Of The Limitations Section:**

3

**Questions For Rebuttal:**

1. How do you label borderline grasps? Edit: reading later now, it sounds like labels are 0-1 probabilities for each grasp succeeding?

2. Dataset scale being necessary for training effective grasp evaluators, re: Why do you distinguish grasp evaluators here? This should generally hold with generative approaches without evaluators as well. In general, it should hold with any grasping approach that trains a neural network.

3. Ln. 119-120: Therefore, y_coll and y_pick are strictly 0 or 1 where y_pgs is a continuous value between 0 and 1?

4. Ln. 129-130: How many trials do you do in IsaacGym per grasp? And is the label computed by a simple <attempts_passed>/<total_attempts> formula?

5. Could you explain what perturbation exactly do you do on grasps likely to pass to solve the label imbalance issue?

6. How are the predefined points on the hand chosen and when? Also how are they defined/represented?

7. Was 5cm chosen as a hyperparam? Was there a reason behind the choice of the number?

8. Ln. 161: “while the hand-object pose did not significantly deviate”. What does this mean? What is hand-object pose in this context? The pose of the hand in reference to the object? Or? Also significantly deviate from what? I imagine a successful grasp where each finger moved by 5cm and the picked object and hand both moved up by 20cm, to have deviated quite a bit from their respective initial configurations.

9. Y_pgs is a logical and between y_coll and y_pick? Therefore still a discrete 0 or 1? Then you add a separate continuous label after averaging 5 trials? Is this label saved separately than y_pgs?

10. Ln. 208-209: "and account for realistic perceptual inputs", re: what does this mean exactly in this context? The dataset is all SIM which vastly differs from real.

11. Could you provide a hypothesis about why the diffusion and fixed samplers, both with an evaluator, perform on par? I’d expect any kind of refinement to perform better if starting from a better initial pose and I assume the fixed set of grasps is strictly a worse initial pose sampler.

12. Why is there no hardware result corresponding to diffusion/point cloud? If a point cloud coming out of e.g. 2-4 views performs on par with NeRF, wouldn’t that just eliminate the need for a tedious data collection as stated in the limitations section?

**Robotics Focus:**

4

**Summary Of Paper:**

The paper introduces a new large scale dataset of simulated grasps that comes with RGB images, point clouds and trained NeRF.

**Summary Of Recommendation:**

Thank you to the authors for this work. The community can definitely benefit from a new large multi-finger grasp dataset that has a wider variety of input modalities. However, further clarity should be provided on how the grasp evaluator argument and experiments connect to the dataset itself. If the argument is that grasp evaluators are generally better than analytic and generative methods on large scale data regimes such as the new dataset, then more comparisons with more baselines from both categories should be added to support this argument. I’m willing to increase my score if more such comparisons or added or if authors explain more clearly the connection of the dataset and the grasp evaluators hypothesis in case I misunderstood.

---

### Official Review · Reviewer_DT3i · 2024-07-18
**data paper, weak accept**

**Originality:** 4
**Technical Quality:** 3
**Clarity Of Presentation:** 4
**Potential Impact:** 3
**Recommendation:** 3
**Confidence:** 4

**Review:**

The introduction outlines the problem through an extensive literature review. The methodology and calculation models are presented clearly and in sufficient detail. The discussion section effectively explains the results obtained. The graphic material is of good quality and self-explanatory.

Strengths:
1. the dataset generation pipeline is clear and easy to follow.
2. The data scale is large and types are rich and can be used in different algorithms.
3. Simulation and real-world tests are implemented in the dataset.
4. The comprehensive ablation studies provide valuable insights into the critical role of learned grasp evaluators,  demonstrating the importance of large-scale datasets for training robust algorithms.

Weaknesses:
1. The dataset generation pipeline is currently designed specifically for the Allegro hand, which may limit its immediate applicability to other dexterous hands.
2. The hardware experiments require a time-consuming data collection process and NeRF training, which may hinder the practical application of the methods.
3. The methods achieve low success rates on certain challenging objects, indicating a need for more diverse training data and improved algorithms.
4. The grasp generation methods are designed for uncluttered tabletop settings, which may not fully represent real-world environments with obstacles and clutter.

**Quality Of The Limitations Section:**

3

**Questions For Rebuttal:**

1. The dataset contains 4.3K objects, but details are missing, how were these items selected?
2. the dataset link in the project website is not available now, when will it be available?
3. The 20 objects in the hardware test are categorized as easy, medium, or hard based on the complexity of their geometry, Is there any quantitative metrics?

**Robotics Focus:**

4

**Summary Of Paper:**

This paper released a new, open-source dataset of 3.5M grasps on 4.3K objects annotated with RGB images, point clouds, and trained NeRFs.  Using this dataset, this paper trained vision-based grasp evaluators that outperform existing analytic and generative modeling-based baselines in both simulated and real-world.

**Summary Of Recommendation:**

This paper released a large scale grasping dataset and generation pipeline, the paper can be recommended as weak accept.

---

### Official Review · Reviewer_JEWj · 2024-07-21
**Simulated Multi-Finger Dataset and Evaluators Enabling Sim2Real Transfer for Grasping**

**Originality:** 4
**Technical Quality:** 4
**Clarity Of Presentation:** 4
**Potential Impact:** 3
**Recommendation:** 3
**Confidence:** 4

**Review:**

### Strengths
- **Large-Scale Multi-Finger Dataset**: The paper introduces a comprehensive dataset for multi-finger grasp evaluation, featuring 3.5 million grasps on 4.3 thousand diverse objects. This extensive dataset includes RGB images, point clouds, success labels, and pre-trained NeRFs, providing a rich source of data for training robust grasp evaluators.
- **Effectiveness in Simulation and Real-World Environments**: The approach demonstrates the effectiveness of the learned evaluators in both simulation and real-world settings, showcasing significant improvements over baseline models and successfully translating to practical applications in robot grasping tasks.
- **Strong Experimental Validation**: The extensive experimental validation, including thorough ablation studies, underscores the robustness and effectiveness of the proposed method. The experiments cover a wide range of scenarios, proving the versatility and practical applicability of the approach in diverse environments.
- **Clear Articulation**: The paper is clearly articulated, with well-structured sections and detailed explanations of the methods and experiments. For example, the authors provide a step-by-step description of the dataset creation process and the implementation of the grasp evaluators, making it easy for readers to understand and replicate the work.

### Weaknesses
- **Limited to Simulated Rigid Objects**: The method and dataset are primarily focused on simulated rigid objects, which limits their applicability to real-world scenarios involving non-rigid or articulated objects.
- **Non-Occluded Single Object Assumption**: The approach assumes the presence of non-occluded single objects, which may not hold true in more complex and cluttered real-world environments.
- **Specific to Allegro Hand**: The dataset generation and evaluation pipeline are specifically designed for the Allegro hand, which may limit the generalizability and applicability of the method to other robotic hands and setups.

**Quality Of The Limitations Section:**

3

**Questions For Rebuttal:**

- In Figure 2 (a), why does fixed methods with learned evaluators perform than that using diffusion? Since the massive improvement comes from the usage of evaluators, could the implementation of evaluators affect the results of fixed vs diffusion?
- What was the runtime and compute resources for collecting this dataset?
- Could the choice of heuristics bias toward certain type of grasps for success grasps?
- What strategies are in place to ensure the robustness of the method when applied to objects with different physical properties (e.g., weight, friction) in real-world scenarios?

**Robotics Focus:**

4

**Summary Of Paper:**

The paper introduces "Get a Grip," a large-scale simulation dataset for multi-finger grasp evaluation, featuring 3.5 million grasps on 4.3 thousand diverse objects annotated with RGB images, point clouds, and NeRFs. This dataset enables the training of robust discriminative grasp evaluators. The approach demonstrates significant improvements in both simulation and real-world environments, showcasing effective sim-to-real transfer. The method outperforms baseline models, highlighting the importance of large-scale datasets and the integration of learned evaluators for robust multi-finger grasping.

**Summary Of Recommendation:**

The paper introduces a large-scale multi-finger grasp evaluation dataset, demonstrating improvements in both simulation and real-world environments. The dataset, featuring 3.5 million grasps on 4.3 thousand diverse objects, is well-annotated with RGB images, point clouds, success labels, and pre-trained NeRFs. This rich data source enables effective training of robust grasp evaluators, facilitating sim-to-real transfer. While the method shows promise, it is primarily focused on simulated rigid objects, assumes non-occluded single objects, and is specific to the Allegro hand, limiting its generalizability to more complex and cluttered real-world scenarios. Given these strengths and limitations, I recommend a weak accept.

---

### Decision · Program_Chairs · 2024-09-04

**Decision:**

Accept

**Comment:**

The authors have provided thorough answers to the questions and concerns of the reviewers. The initial scores ( 2 weak accepts and 1 weak reject) have been turned into 3 weak accepts.